# Surface exclusion of IncC conjugative plasmids and their relatives

Nicolas Rivard[1¤a], Malika Humbert[1☉], Kévin T. Huguet[1☉¤b], Aurélien Fauconnier[1☉], César Pérez Bucio[2], Eve Quirion[1], Vincent Burrus[1]*

1 Département de biologie, Université de Sherbrooke, Sherbrooke, Quebec, Canada, 2 Instituto Tecnológico y de Estudios Superiores de Monterrey, Monterrey, Nuevo León, Mexico

☉ These authors contributed equally to this work.
¤a Current address: Laboratoire d'Ingénierie des Systèmes Macromoléculaires, UMR7255, Institut de Microbiologie de la Méditerranée, Aix-Marseille Univ, CNRS, Marseille, France.
¤b Current address: CNRS, LCPME, UMR 7564, Université de Lorraine, Villers-lès-Nancy, France
* Vincent.Burrus@USherbrooke.ca

**Data Availability Statement:** All relevant data are within the manuscript and its Supporting Information files. Data sets for graphs are available in S1 File. Raw sequencing data of TraDIS experiments were submitted to Genbank Sequence

## Abstract

The phenomenon of exclusion allows conjugative plasmids to selectively impede the entry of identical or related elements into their host cell to prevent the resulting instability. Entry exclusion blocks DNA translocation into the recipient cell, whereas surface exclusion destabilizes the mating pair. IncC conjugative plasmids largely contribute to the dissemination of antibiotic-resistance genes in Gammaproteobacteria. IncC plasmids are known to exert exclusion against their relatives, including IncC and IncA plasmids, yet the entry exclusion factor *eexC* alone does not account for the totality of the exclusion phenotype. In this study, a transposon-directed insertion sequencing approach identified *sfx* as necessary and sufficient for the remaining exclusion phenotype. Sfx is an exclusion factor unrelated to the ones described to date. A cell fractionation assay localized Sfx in the outer membrane. Reverse transcription PCR and beta-galactosidase experiments showed that *sfx* is expressed constitutively at a higher level than *eexC*. A search in Gammaproteobacteria genomes identified Sfx homologs encoded by IncC, IncA and related, untyped conjugative plasmids and an uncharacterized family of integrative and mobilizable elements that likely rely on IncC plasmids for their mobility. Mating assays demonstrated that *sfx* is not required in the donor for exclusion, ruling out Sfx as the exclusion target. Instead, complementation assays revealed that the putative adhesin TraN in the donor mediates the specificity of surface exclusion. Mating assays with TraN homologs from related untyped plasmids from *Aeromonas* spp. and *Photobacterium damselae* identified two surface exclusion groups, with each Sfx being specific of TraN homologs from the same group. Together, these results allow us to better understand the apparent incompatibility between IncA and IncC plasmids and to propose a mechanistic model for surface exclusion mediated by Sfx in IncC plasmids and related elements, with implications for the rampant dissemination of antibiotic resistance.

Read Archive (SRA) under BioProject accession number PRJNA857050 with the following BioSample accession numbers: SAMN32907975 and SAMN32907976 for E. coli BW25113 Nx bearing pVCR94Kn ΔeexC::cat as the recipient strain (input); SAMN32907978 and SAMN32907979 for E. coli BW25113 Nx transconjugants bearing pVCR94Kn ΔeexC::cat and pSU4628 (output). Complete data from aligned reads can also be visualized using the UCSC genome browser at: https://genome.ucsc.edu/cgi-bin/hgHubConnect?hgHub_do_redirect=on&hgHubConnect.remakeTrackHub=on&hgHub_do_firstDb=1&hubUrl=https://datahub-101-cw2.p.genap.ca/Rivard_2023/pVCR94dX3deexC.hub.txt.

**Funding:** This work was supported by Discovery Grant (RGPIN-2021-02814) from the Natural Sciences and Engineering Council of Canada (NSERC, https://www.nserccrsng.gc.ca/) and Project Grants (PJT-153071 and PJT-186081) from the Canadian Institutes of Health Research (CIHR, https://cihr-irsc.gc.ca/e/) to V.B. N.R. is the recipient of an Alexander Graham Bell Canada Graduate Scholarship from NSERC. C.P.B. is the recipient of a Globalink Research Internship from Mitacs. The funders had no role in study design, data collection and analysis, decision to publish, or preparation of the manuscript.

**Competing interests:** I have read the journal's policy and the authors of this manuscript have the following competing interests: V.B. has served as guest editor in 2021/2022 for PLoS Genetics. All other authors have no competing interest.

## Author summary

Bacterial conjugation plays a pivotal role in the evolution of bacterial populations. The circulation of drug resistance genes bolsters the emergence of multidrug-resistant pathogens, with which contemporary medicine struggles to cope. Exclusion is a natural process preventing the redundant acquisition of a plasmid via conjugation by a host harbouring an identical or similar plasmid. Although exclusion has been known for the past half-century, the mechanisms involved remain poorly understood. This study describes an exclusion factor, Sfx, encoded by IncC, IncA and related conjugative plasmids and by unrelated integrative and mobilizable elements. We report that Sfx is a lipoprotein of the recipient that selectively inhibits conjugation based on the adhesin TraN expressed at the surface of the donor. We propose a mechanistic model for Sfx-mediated exclusion. Ultimately, a better understanding of exclusion could facilitate the design of conjugation inhibitors targeting mating pair formation to curb the circulation of drug-resistance genes in healthcare settings, agriculture, animal husbandry and food and drug production.

## Introduction

IncC plasmids are large conjugative plasmids frequently associated with multidrug resistance phenotypes in a broad range of Gammaproteobacteria species [1]. IncC plasmids' ability to mobilize non-autonomous, unrelated integrative and mobilizable elements (IMEs) of the *Salmonella* Genomic Island 1 (SGI1) and MGI*Vch*Hai6 families exacerbates their importance in antibiotic resistance dissemination [2–5]. SGI1 and its multiple variants are important vehicles of antibiotic resistance genes and are frequently found in *Salmonella enterica* and a broad range of Gammaproteobacteria [6]. MGI*Vch*Hai6 conferred multidrug resistance to *Vibrio cholerae* non-O1/non O139 infecting cholera patients during the 2010 cholera outbreak in Haiti [4]. IncC, IncA and untyped IncA/C-like (ACL) plasmids (e.g., pAsa4c, pAhD4-1, pAQU1) share a syntenic set of genes involved in conjugation, DNA repair, and regulation. All seem regulated by closely related homologs of the master activator of transfer AcaCD [7–9]. IncC plasmid-encoded AcaCD promotes the excision and mobilization of MGI*Vch*Hai6-like IMEs [4,5]. AcaCD also stimulates the excision and replication of SGI1 and the expression of its *traN*, *traG* and *traH* genes, which modify the mating apparatus of IncC plasmids to enhance its dissemination [7,10,11].

Large conjugative plasmids are autonomous replicons that usually remain at a low-copy number. The entry into the same cell of a plasmid sharing similar replication or partitioning determinants promotes plasmid destabilization and loss, a phenomenon known as incompatibility [12]. Exclusion refers to the mechanisms employed by conjugative elements to preclude instability resulting from incompatibility by hindering redundant transfer into the cells they reside [13]. While the definite nature of exclusion mechanisms remains elusive, they can be categorized as entry and surface exclusion by whether they interfere with DNA transfer or mating pair stabilization [13,14]. Exclusion seems to be a staple of the conjugative plasmid lifestyle. Exclusion factors have been identified in the IncF, IncHI1, IncI, IncN, IncP, IncW and ColE1 families, plasmids of the pKPC_UVA01 replicon type and the enterococcal conjugative plasmid pCF10 [14–22]. Integrative and conjugative elements (ICEs) of the SXT/R391 family and ICE*Bs*1 of *Bacillus subtilis* also encode exclusion determinants [23,24]. Most exclusion factors identified to date are associated with the cytoplasmic membrane, where they would exert entry exclusion, with only the IncF TraT firmly established as a surface exclusion factor [13,14]. Like TraT, the IncHI1 EexB localizes in the outer membrane and is thought to engage

in surface exclusion [15]. Likewise, the surface-exposed PrgA protease of pCF10 reduces aggregation and transfer [22].

Exclusion is not specific to the transferred DNA [25], and proteins in the donor have been identified as targets of exclusion factors in several systems. EexA and EexB of IncHI1 plasmids and TrbK of pKPC_UVA01 appear to act concurrently as exclusion factors and exclusion targets, as they are necessary in both the donor and recipient for exclusion to take place [15,21]. Other exclusion factors target different plasmid-encoded proteins. The IncI ExcA targets TraY, which is found in the cytoplasmic membrane of the donor [16]. Similarly, the IncF TraS and SXT/R391 Eex of the recipient seem to interact directly with the VirB6 homolog TraG of the donor, and the regions involved in both proteins are cytoplasmic [23,26]. In the cases above, the genes encoding the exclusion factor and its target are adjacent. In IncP, IncW and IncN plasmids, for which the exclusion target remains unknown, the exclusion gene abuts a gene coding for a VirB6 homolog, suggesting a widespread usage of this target beyond the IncF and SXT/R391 families [13].

The surface exclusion factor TraT of IncF plasmids reduces the percentage of mating aggregates, and five contiguous residues determine specificity between exclusion groups [14,27]. Harrison *et al.* proposed that TraT destabilizes the mating pair by interacting with the pilus tip; however, it was eventually shown that the pilin does not define exclusion specificity [27,28]. The discovery of interactions between TraT and the outer membrane protein OmpA in the recipient and between OmpA in the recipient and the putative adhesin TraN in the donor suggested surface exclusion could instead result from TraT preventing docking of TraN to OmpA, thus hindering mating pair stabilization [29–31]. The identification of TraN receptor specificity groups across IncF plasmids makes this hypothesis extremely appealing, yet *traN* substitutions failed to flip exclusion specificity between F and R100-1 [30,32,33].

IncC plasmids share with IncA plasmids the same entry exclusion system, *eexC*/*traG*, and inhibit each other's entry into the cell in which they reside [9,34]. During the characterization of *eexC*/*traG*, we reported that knocking-out *eexC* of the IncC plasmid pVCR94 in the recipient only reduced the exclusion phenotype in mating assays using an IncC$^+$ donor. Likewise, the deletion of a 45.6-kb fragment encompassing 33 open reading frames located between *traN* and *traF* led to a statistically significant reduction of exclusion. Hence, another mechanism likely prevents redundant transfer between cells containing identical plasmids.

In this report, we investigate the cause of the incomplete abolition of exclusion by a recipient strain bearing an IncC plasmid lacking the entry exclusion gene *eexC*. Using a transposon-directed insertion sequencing (TraDIS) approach, we identified IncC-borne candidate genes whose disruption in recipient cells facilitated the acquisition of an incoming mobilizable plasmid from an IncC$^+$ donor. Further analyses revealed that a single open reading frame, hereafter named *sfx*, flanked by *traN* and *acaB*, accounts for the residual exclusion phenotype. Deletion of *sfx* with *eexC* completely abolished exclusion. Signal peptide prediction and a subcellular localization assay suggest the Sfx protein is an outer-membrane lipoprotein and support a role as a surface exclusion factor. Sfx belongs to an uncharacterized family of broadly distributed surface exclusion proteins encoded by IncC plasmids, their relatives and several putative IMEs. Surface exclusion assays designed to test *sfx*/*traN* gene pairs from ACL plasmids revealed that *sfx* and *traN* are part of a surface exclusion system, enabling us to propose a model of surface exclusion of ACL plasmids.

## Results

### *vcrx085* impedes conjugative transfer between IncC$^+$ cells

To identify additional factors responsible for the inhibition of transfer between IncC$^+$ cells, we used transposon-directed insertion sequencing (TraDIS) aimed at mapping Tn*5* insertions

that enhance the entry of the mobilizable plasmid pSU4628 (pClo) transferred from an IncC+ donor strain into a Tn5+ IncC+ recipient library (Fig 1A). pClo was used as a proxy for conjugative transfer to bypass IncC plasmid incompatibility. Briefly, we constructed a high-density mini-Tn5 (Sp$^r$) insertion library in *E. coli* GG56 carrying pVCR94$^{Kn}$ Δ*eexC*::*cat*, a kanamycin-resistant variant of pVCR94 unable to exert entry exclusion in recipient cells [9]. This set of mutants formed the input library containing Tn5 insertions in the chromosome or pVCR94 that allowed plasmid replication and maintenance. The input library was then used as the recipient in a mating assay with a donor strain carrying pVCR94$^{Kn}$ and the mobilizable plasmid pClo. GG56 transconjugant colonies that acquired pClo made up the output library. We expected that the genes or sequences of pVCR94$^{Kn}$ Δ*eexC*::*cat* more frequently disrupted in the output than in the input (high insertion index ratio) were those impeding the mobilization of pClo between IncC+ cells. Since our assays monitored pClo entry only and aimed to identify plasmid-borne exclusion factors, we did not consider the role of chromosomal genes in the recipient in this study.

We identified nine candidate genes in pVCR94$^{Kn}$ Δ*eexC*::*cat* with insertion index ratios above the arbitrary threshold of 5 (Table 1 and Figs 1B, 1C, and S1). Among these genes, only *acr2* (*vcrx150*) is a known repressor of conjugative transfer [7]. To validate the ability of the candidates to inhibit incoming conjugative transfer, we tested mobilization of pClo into the corresponding deletion mutants of pVCR94$^{Sp}$ used as recipients. All but one had no impact on transfer of pClo (Fig 1D). The deletion of *vcrx085* (hereafter referred to as *sfx* for surface exclusion) improved conjugation to the same level similar as the deletion of the entry exclusion factor *eexC*. Like *eexC*, we could complement the *sfx* deletion in *trans* (Fig 1D, pAH56). Both mutations combined completely abolished exclusion, allowing conjugation to a level comparable to an empty recipient. Conversely, the standalone expression of *sfx* and *eexC* from their native promoter in the absence of an IncC plasmid in the recipient matched the full exclusion phenotype of the wildtype IncC plasmid (Fig 1D, pBeloBAC11). Deletion of *eexC*, *sfx*, or both slightly improved the fitness of the cells (increased CFU counts/ml) compared to wild-type pVCR94$^{Sp}$, restoring the same level as empty cells (S2 Fig). Conversely, the complementation of *eexC* or *sfx* from pBAD30 slightly decreased the fitness. These observations suggest that *eexC* and *sfx* affect the growth of cells bearing pVCR94$^{Sp}$.

## Sfx is a constitutively expressed membrane protein

The SignalP 6.0 server predicts a lipoprotein signal peptide in the translation product of *sfx* (S3A Fig), suggesting Sfx is displayed on the cell surface [35]. A C-terminally 3xFLAG-tagged version of *sfx* was cloned under the control of the inducible promoter $P_{tac}$. The Sfx$^{3xFLAG}$ polypeptide has a predicted size of 33.4 kDa before signal peptide cleavage and 31.7 kDa afterwards. The ability of Sfx$^{3xFLAG}$ to mediate surface exclusion was confirmed in a conjugation assay and showed only a twofold reduction of exclusion compared to untagged Sfx (S4 Fig). A cell fractionation assay confirmed the presence of Sfx$^{3xFLAG}$ in the inner and outer membrane fractions (Fig 2A). In contrast, the known outer membrane protein OmpC, used as a control, was only detected in the outer membrane fraction. This cellular localization remains consistent with the role of Sfx in surface exclusion. The accumulation of Sfx$^{3xFLAG}$ in the inner membrane suggests that the high expression level of *sfx*$^{3xFLAG}$ from $P_{tac}$ on a multi-copy vector saturates the Lpt trafficking pathway. We could not undoubtedly assign the processed and unprocessed forms in the inner membrane fraction, and two additional products appeared between 20 and 25 kDa on the western blot. These weaker bands are presumably degradation products of an unknown nature.

*sfx* is located downstream of the transfer activator *acaB*, which is expressed from an AcaCD-inducible promoter (Fig 2B) [37]. Hence, we wondered whether *sfx* expression was

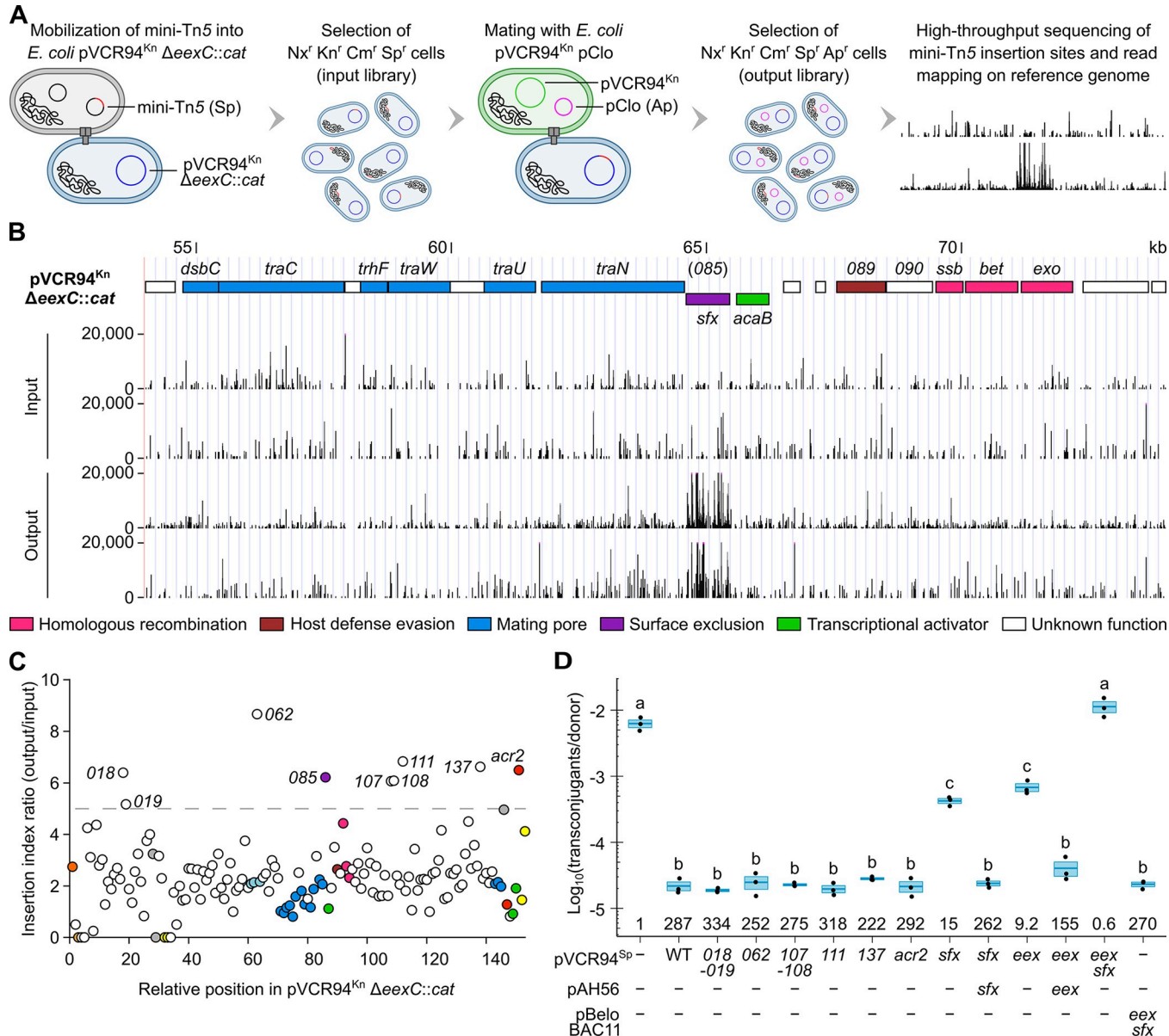

**Fig 1. Identification of an additional IncC exclusion factor.** (A) Overview of the TraDIS workflow to identify exclusion factors in pVCR94$^{Kn}$ $\Delta eexC::cat$. (B) Mini-Tn*5* reads near *vcrx085*. The tracks plot the number of reads from two independent replicates as a function of position in pVCR94$^{Kn}$ $\Delta eexC::cat$ for both the input and output libraries. ORFs with similar functions are colour-coded as indicated in the panel. This panel was created using the UCSC Genome Browser (http://genome.ucsc.edu). (C) Nine loci of unknown function show a mini-Tn*5* insertion index ratio greater than 5 (dashed line), calculated as the ratio of insertion counts between the output and input libraries. ORFs with similar functions are colour-coded as indicated in S1 Fig. (D) *sfx* (*vcrx085*) is the only functional exclusion factor among candidates identified in (B) and suffices alongside *eexC* (*eex*) to account for the totality of IncC exclusion. *E. coli* VB112 (Rf$^r$) containing pVCR94$^{Kn}$ and pClo (Ap$^r$) served as the donor strain, and *E. coli* GG56 (Nx$^r$) containing the specified derivatives of pVCR94$^{Sp}$ served as the recipient strains. When indicated, *eexC* or *sfx* were expressed in the recipients from single-copy, chromosomally integrated pAH56 (Kn$^r$) in the presence of IPTG or together from pBeloBAC11 (Cm$^r$) under the control of their respective native promoters. Transconjugants containing pClo were selected as the Nx$^r$ Ap$^r$ colonies. Crossbars show the mean and standard error of the mean of three independent experiments. One-way ANOVA (p = 1.6e-25) with a Tukey-Kramer post-test was used on the log$_{10}$-transformed values to compare the means. Statistical significance (S2 File) is shown as a compact letter display for pairwise comparisons where means grouped under identical letters are not statistically different. Exclusion indices, shown at the bottom of each crossbar, are calculated as the frequency of transfer into the empty recipient divided by the frequency of transfer into the indicated mutant.

**Table 1. Candidate genes identified by TraDIS.**

| ORF | Fold change | Predicted size (aa) | Predicted product[a] |
|---|---|---|---|
| *vcrx018* | 6.39 | 119 | Hypothetical protein |
| *vcrx019* | 5.17 | 148 | $Fe^{3+}$-siderophore ABC transporter permease |
| *vcrx062* | 8.67 | 194 | Hypothetical protein |
| *vcrx085 (sfx)* | 6.22 | 286 | Lipoprotein (Sec/SPII) |
| *vcrx107* | 6.07 | 114 | Hypothetical protein |
| *vcrx108* | 6.08 | 100 | Hypothetical protein |
| *vcrx111* | 6.83 | 184 | Hypothetical protein |
| *vcrx137* | 6.63 | 163 | Hypothetical protein |
| *acr2* | 6.50 | 139 | Transcriptional repressor |

[a]As predicted by NCBI blastp or SignalP 6.0.

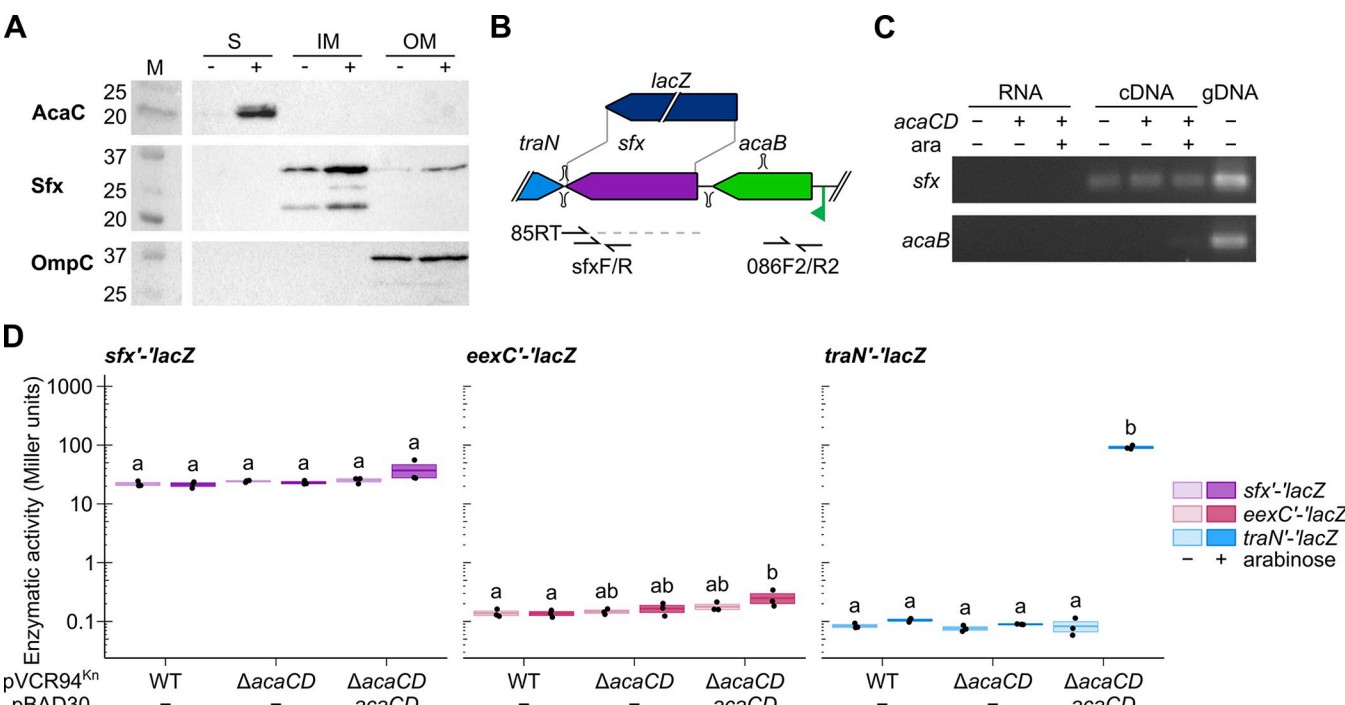

**Fig 2. Expression of IncC surface exclusion is independent of the master activator of transfer AcaCD.** (A) Cell fractionation assay. The soluble (S), inner membrane (IM), and outer membrane (OM) cell fractions of *E. coli* DH5α expressing 3xFLAG-tagged $Sfx_{94}$ or 3xFLAG-tagged AcaC from pAH56 under the control of the IPTG-inducible $P_{tac}$ promoter with (+) and without (-) IPTG were analyzed by SDS-PAGE and immunodetected using rabbit anti-FLAG or anti-OmpC antibodies. Molecular weight markers (M) are indicated on the left. A vertical white line separates non-adjacent lanes from a single blot. AcaC-3xFLAG (22 kDa), a known cytoplasmic protein [7] and OmpC (36 kDa), a known outer membrane protein [36], were used as controls in the fractionation. (B) Schematic representation of the *acaB-sfx* locus. The relative positions of reverse transcription primer 85RT and primers used to amplify *sfx* and *acaB* are shown. A dotted line shows the reverse transcription product. Stem loops represent predicted Rho-independent terminators. The translational *lacZ* fusion used for enzymatic assays was introduced after the third codon of *sfx*. (C) *sfx* is expressed constitutively and independently of the upstream AcaCD-dependent $P_{acaB}$ promoter. A 2% agarose gel of the PCR products from an *sfx*-derived reverse transcribed RNA. Untreated RNA (RNA) and genomic DNA from *E. coli* GG56 (Nx[r]) containing pVCR94[Kn] Δ*acaCD* (gDNA) were used as negative and positive controls, respectively. (D) β-galactosidase activity of GG56 (Nx[r]) containing pVCR94[Kn] with the *eexC'-'lacZ*, *sfx'-'lacZ*, or *traN'-'lacZ* translational fusions, with or without p*acaCD* and arabinose. Crossbars show the mean and standard error of the mean of three independent experiments. One-way ANOVA (*sfx*, p = 0.06; *eex*, p = 0.037; *traN*, p = 2.6e-15) with a Tukey-Kramer post-test was used to compare the means. Statistical significance is shown as a compact letter display for pairwise comparisons where means grouped under identical letters are not statistically different.

controlled by AcaCD. To test this hypothesis, we constructed a translational fusion between the third codon of *sfx* and the eighth codon of *lacZ* in pVCR94$^{Kn}$. Expression of AcaCD did not impact the β-galactosidase activity of the *sfx'-'lacZ* fusion. A similarly constructed *eexC'-'lacZ* fusion yielded a ~150-fold lower β-galactosidase activity, regardless of the presence of AcaCD (Fig 2D). We used a similar *traN'-'lacZ* fusion as a control and observed a very low level of expression in IncC$^+$ cells, which increased ~260 fold upon AcaCD overexpression. Furthermore, to test whether *sfx* and *acaB* are part of the same transcript, we attempted to amplify both loci from a cDNA synthesized from the 3' end of *sfx*. While *sfx* could be amplified regardless of AcaCD, *acaB* remained undetectable, suggesting that these two genes are expressed as independent mRNA transcripts (Fig 2C). A Rho-independent terminator predicted between *sfx* and *acaB* supports this idea (Figs 2B and S3B). Altogether, our results indicate that *sfx* is constitutively expressed independently of AcaCD at a much higher level than *eexC*.

## Genes encoding homologs of Sfx$_{94}$ are found in diverse genomic backgrounds

To explore the diversity of exclusion factors, we first used NCBI tblastn to identify Sfx$_{94}$ homologs in the ACL plasmids pAsa4c of *Aeromonas salmonicida*, pAhD4-1 of *Aeromonas hydrophila*, and pAQU1 of *Photobacterium damselae* that are known untyped relatives of IncA and IncC plasmids [8,9]. For pAsa4c and pAhD4-1, closely related homologs were found adjacent to the gene *traN*. For pAQU1, the search resulted in a single hit (*215*, hereafter referred to as *sfx$_{AQU1}$*) whose product shares only 39% identity over 273 amino acid residues with Sfx$_{94}$. *sfx$_{AQU1}$* is surrounded by genes of unknown function. Sfx$_{AQU1}$ was manually added to the dataset for downstream analyses. We next searched the GenBank database for homologs of Sfx$_{94}$ and extracted the resulting sequences for further analyses. 130 unique proteins were identified (S3 File), most encoded by ACL plasmids (Fig 3A). 78 Sfx homologs share more than 90% identity with Sfx$_{94}$, including nine variants encoded by IncC plasmids. All the Sfx homologs encoded by IncC plasmids are inside this cluster. Sfx$_{94}$ belongs to an identical protein group of 193 representatives, all encoded by IncC or putative IncC plasmids. Sfx$_{RA1}$ from the IncA plasmid pRA1 belongs to the Sfx$_{94}$ cluster as they share 92% identity over 286 amino acid residues. All plasmids outside the Sfx$_{94}$ cluster are ACL plasmids, including pAsa4c and pAhD4-1. In most ACL plasmids and with the rare exception of pAQU1, *sfx* lies downstream of the *dsbC-traC-trhF-traWU* operon between *traN* and the transfer activator gene *acaB* (Fig 3C).

The remaining Sfx proteins are encoded by putative IMEs integrated at four distinct chromosomal locations. First, ten of these IMEs belong to the MGI*Vch*Hai6 family and are integrated at the 5' end of *dusA* (tRNA-dihydrouridine synthase A) in environmental and clinical *Aeromonas* strains [5]. Represented by IE*Ave*Tha1 (Fig 3C), all possess MGI*Vch*Hai6-like *mobI* and *oriT* and predicted AcaCD-responsive promoters (S2 Table). A second group represented by IE*Vch*Aut1 comprises six IMEs integrated at the 3' end of *yicC* (putative RNase adaptor protein YicC) in environmental and clinical *Vibrio* strains. In addition to a putative recombination directionality factor related to RdfM of MGI*Vfl*Ind1 [38], these IMEs carry a set of conserved genes, including a predicted type IV coupling protein (*traD*) and a putative relaxase of the MOB$_F$ family (*traI*). Finally, two IMEs coding for an identical Sfx are integrated at the 3' end of *tRNA-TRP* and the 5' end of *rlmD* (23S rRNA uracil methyltransferase) in environmental isolates of *Shewanella chilikensis* and *Shewanella putrefaciens*, respectively. Despite sharing a conserved core set of genes with IE*Vch*Aut1, the Sfx protein of these two IMEs resembles and clusters with Sfx$_{94}$ (Fig 3A). Although we can predict AcaCD binding sites across these IMEs (S2 Table), no other conserved features of the SGI1 and MGI*Vch*Hai6

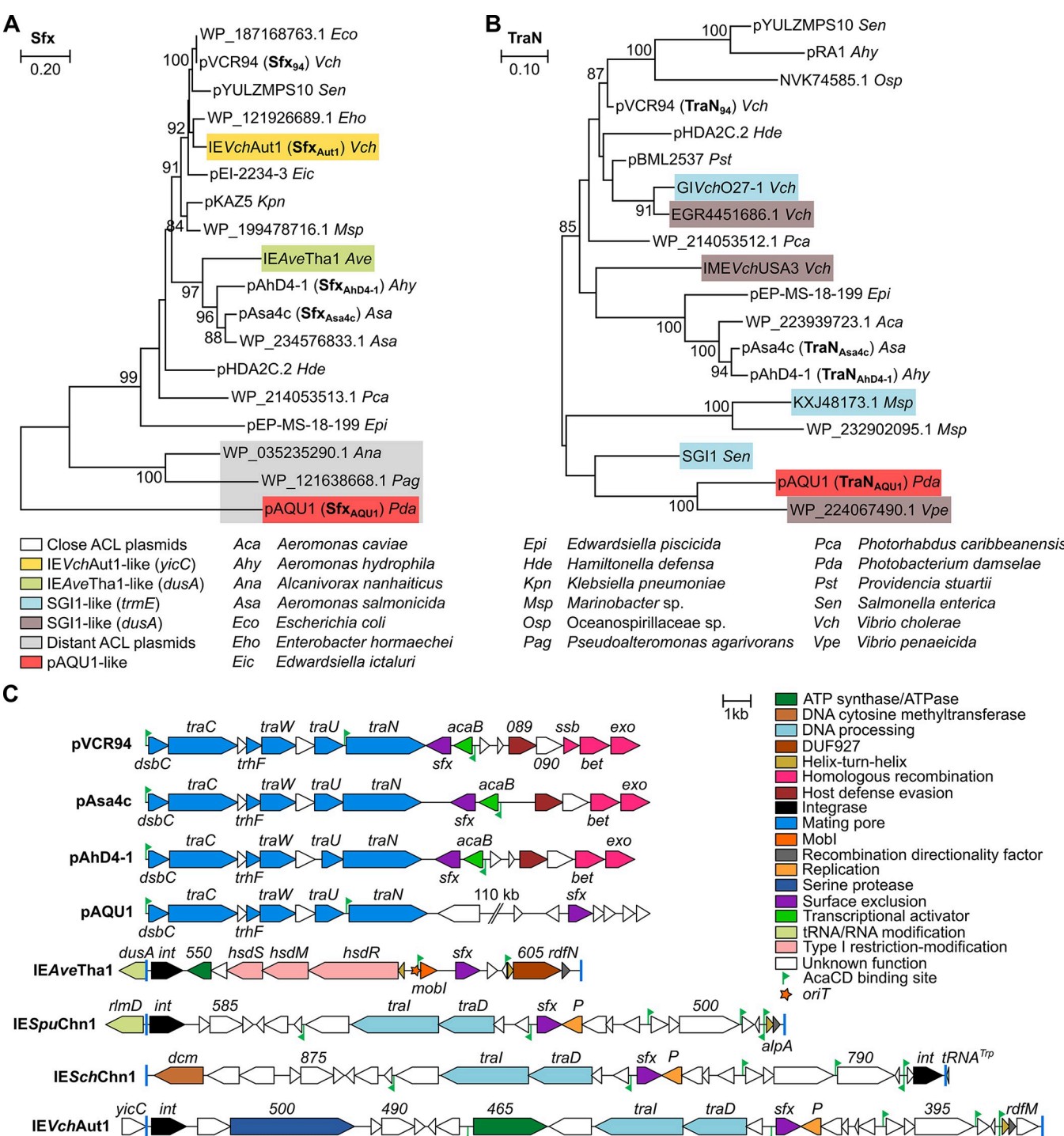

**Fig 3. Sfx homologs are found across a wide range of Gammaproteobacteria.** Maximum likelihood phylogenetic analyses of Sfx (A) and TraN (B) homologs. Trees with the highest likelihood (9,621.78 and 28,204.23 for Sfx and TraN, respectively) are shown. Bootstrap supports are indicated as percentages at the branching points only when >80%. Branch lengths represent the number of substitutions per site over 269 and 813 amino acid positions for Sfx and TraN, respectively. Only one representative per cluster of similar proteins (>90%) is shown in each tree, except for TraN$_{Asa4c}$ and TraN$_{AhD4-1}$, which share 95% identity over the aligned sequences. S3 File provides additional details on host strains. (C) Schematic representation of the genetic context of *sfx* homologs in a selection of putative IMEs and ACL plasmids. Predicted *attL* and *attR* attachment sites flanking the IMEs are represented by blue bars. pAQU1 encodes homologs of *acaB*, *vcrx087-088-089-090*, *bet* and *exo* clustered in a distinct locus not shown here. ORFs with similar functions are colour-coded as indicated in the panel.

families was found, suggesting they belong to an unrelated IME family mobilizable by IncC plasmids.

## Surface exclusion specificity is determined by TraN

To test the role of Sfx in the donor, we monitored the transfer of pClo from donors carrying the $\Delta eexC$ and $\Delta sfx$ deletion mutants of pVCR94$^{Sp}$ toward an IncC$^+$ strain (pVCR94$^{Kn}$). No alleviation of exclusion, which would have increased the transfer frequency, was observed (Fig 4A). On the contrary, a modest yet significant decrease was observed for the $\Delta sfx$ and $\Delta eexC \Delta sfx$ mutants. This phenotype might result from augmented transfer between donors, which is likely detrimental to efficient transfer toward recipients. Hence, $sfx$ is not self-targeting to promote surface exclusion.

Next, we focused on TraN as a potential target for surface exclusion, as the gene $traN$ encodes a putative adhesin thought to stabilize the mating pair, and it is adjacent to $sfx$ in most conjugative plasmids found but pAQU1 [30,31,39]. A search for homologs of TraN$_{94}$ yielded 412 homologous proteins (S3 File), most encoded by ACL plasmids. Others belonged to IncC-mobilizable IMEs of the SGI1 family integrated into $trmE$ or $dusA$, with no clear relationship between TraN phylogeny and insertion site (Fig 3B). The cluster represented by pAQU1 differs significantly from the close ACL plasmids as the $dsbC$-$traC$-$trhF$-$traWUN$ and $acaB/bet$-$exo$ regions are not adjacent, and $sfx$ lies elsewhere.

To investigate the involvement of TraN in Sfx-dependent exclusion, we used in mating assays a $traN$ deletion mutant of pVCR94$^{Sp}$ complemented ectopically with either its native $traN_{94}$ or the distantly related homolog $traN_{AQU1}$ [40]. Despite sharing only 66% identity with TraN$_{94}$, TraN$_{AQU1}$ fully complemented the $\Delta traN_{94}$ mutation (Fig 4B). Furthermore, whereas the expression of $sfx_{94}$ in the recipient strongly inhibited conjugation supported by $traN_{94}$, it did not impair transfer supported by $traN_{AQU1}$ (Fig 4B). This result implies that TraN is, if not a direct interactant of Sfx, at least a key player in Sfx-mediated exclusion and the determinant

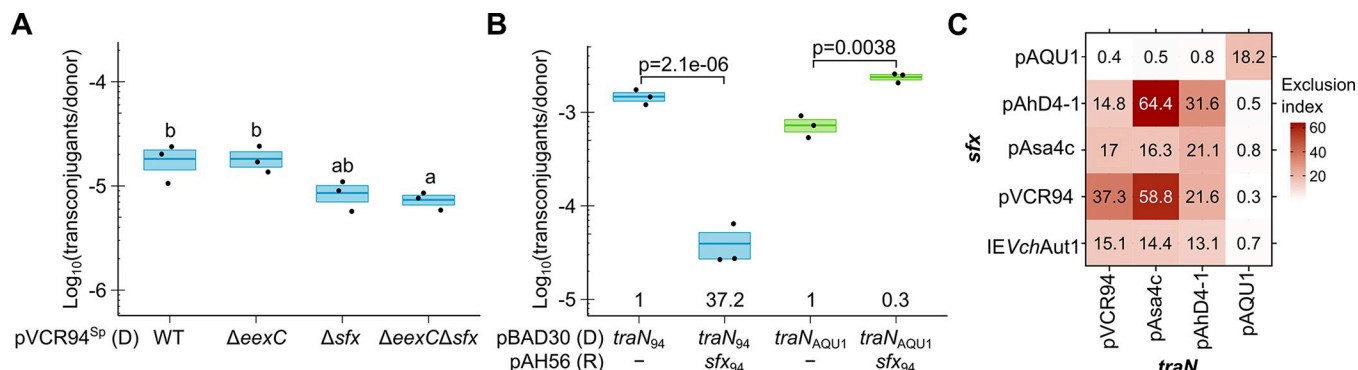

**Fig 4. TraN directs surface exclusion specificity.** (A) The absence of $eexC$ and $sfx$ in the donor decreases the frequency of transconjugant formation. pClo (Ap$^r$) was mobilized from *E. coli* VB112 (Rf$^r$) bearing the indicated deletion mutants of pVCR94$^{Sp}$ (donor, D) into *E. coli* GG56 (Nx$^r$) bearing pVCR94$^{Kn}$ (recipient). Transconjugants containing pClo were selected as the Nx$^r$ Ap$^r$ colonies. Crossbars show the mean and standard error of the mean of three independent experiments. One-way ANOVA (p = 0.016) with a Tukey-Kramer post-test was used on the log$_{10}$-transformed values to compare the means. Statistical significance (S2 File) is shown as a compact letter display for pairwise comparisons where means grouped under identical letters are not statistically different. (B) $sfx_{94}$ is ineffective against the $traN_{94}$ distant homolog $traN_{AQU1}$. Pairwise t-tests were used on the log$_{10}$-transformed values to compare the means. Exclusion indices are calculated by dividing the frequency of transfer of pVCR94$^{Sp}$ $\Delta traN$ to the empty recipient divided by the transfer frequency to the recipient expressing $sfx_{94}$. (C) Heatmap of surface exclusion indices of $sfx$ and $traN$ pairs from a diverse set of ACL plasmids. Exclusion indices are calculated by dividing the frequency of transfer of pVCR94$^{Sp}$ $\Delta traN$ to the empty recipient divided by the transfer frequency to the recipient expressing the indicated $sfx$ variant. In (B) and (C), *E. coli* GG56 (Nx$^r$) containing pVCR94$^{Sp}$ $\Delta traN$ and expressing the indicated $traN$ variant from pBAD30 (Ap$^r$) (donor, D) was mated with *E. coli* CAG18439 (Tc$^r$) expressing the indicated $sfx$ variant from a single-copy, chromosomally integrated pAH56 (Kn$^r$) (recipient, R). Transconjugants containing pVCR94$^{Sp}$ $\Delta traN$ were selected as the Tc$^r$ Sp$^r$ colonies.

of exclusion specificity. To further investigate the diversity of exclusion specificity, we tested the *traN*/*sfx* pairs of pAsa4c and pAhD4-1, two plasmids belonging to distinct entry exclusion groups, and pAQU1 [9,41,42]. The corresponding genes were tested in pairs in all possible combinations to determine exclusion indices. Two separate exclusion groups emerged (Figs 4C and S5). *sfx* homologs from pVCR94, pAsa4c and pAhD4-1 specifically exerted exclusion against *traN* homologs from these same plasmids. $sfx_{AQU1}$ only excluded $traN_{AQU1}$. In addition, we found that $sfx_{Aut1}$ from the putative IME IE*Vch*Aut1 is functional. $sfx_{Aut1}$ exhibits a surface exclusion pattern resembling $sfx_{Asa4c}$, excluding transfer mediated by *traN* of pVCR94, pAsa4c and pAhD4-1 but not pAQU1.

## Discussion

All known conjugative plasmids encode at least one exclusion factor [13]. An entry exclusion system involving the entry exclusion factor EexC and the VirB6-homolog TraG was previously identified in IncA and IncC conjugative plasmids [9]. Here, we identified a second exclusion factor within a region suspected by Humbert *et al.* to be responsible for additional exclusion activity [9]. A TraDIS experiment pointed to nine candidate genes likely to enhance incoming transfer once disrupted. *sfx* was the only one whose activity in exclusion could be experimentally validated. Together, *eexC* and *sfx* account for the totality of the exclusion phenotype (Fig 1D), akin to the exclusion mediated by *traS* and *traT* of the F plasmid [43]. We could not detect any impact of abolishing exclusion on cell viability (S2 Fig). This result strongly contrasts with reports on F plasmid exclusion for which double mutants of *traS* and *traT* could not be isolated [43]. This discrepancy could be due to the strong repression of the conjugative machinery of IncC plasmids (Fig 2D), whereas F transfer is constitutive due to the inactivation of F *finO* by IS*3* [7,44].

The translation product of *sfx* is a predicted lipoprotein, and we detected Sfx in the outer membrane (Fig 2A), a trait shared with the surface exclusion factor TraT of the F plasmid [14,45]. We also detected Sfx in the inner membrane, which could result from the overproduction of the tagged protein from the strong promoter. Further characterization of Sfx and its subcellular localization is ongoing to understand better the mechanism of action of surface exclusion. Nevertheless, we showed that exclusion mediated by *sfx* is inhibited or abolished by the substitution of the putative adhesin TraN (Fig 4C), suggesting that exclusion specificity is determined by *sfx*/*traN* pairs. Altogether, our observations support a mechanism of surface exclusion.

Our search in the Genbank database using $Sfx_{94}$ failed to reveal any known entry or surface exclusion factor, including F plasmid TraS or TraT, establishing Sfx of IncA and IncC plasmids as an unrelated surface exclusion factor (S3 File). As the ACL plasmids pAsa4c and pAhD4-1 belong to the same surface exclusion group as pVCR94 despite their divergence (Fig 4C), IncA and IncC plasmids are expected to exert surface exclusion against each other, hence contributing alongside entry exclusion to the previously observed exclusion phenotype between the two incompatibility groups [9,34]. Among the Sfx homologs tested in this study, only $Sfx_{AQU1}$, which failed detection by the NCBI blastp search due to low similarity with $Sfx_{94}$, belongs to a separate surface exclusion group (Fig 4C). A pairwise comparison illustrates that, like $Sfx_{AQU1}$, the putative surface exclusion proteins WP_035235290.1 and WP_121638668.1 share minimal homology with all other representatives (Figs 3A and S6). Therefore, we speculate that neither belongs to the two surface exclusion groups described here and that all other IncC, IncA and close ACL plasmids belong to the $Sfx_{94}$ surface exclusion group. This hypothesis is further supported by the exclusion pattern of $sfx_{Aut1}$ (IE*Vch*Aut1), virtually identical to $sfx_{Asa4c}$. From our observations with ACL plasmids, surface exclusion strikingly contrasts with entry exclusion,

which seems much more specific. Indeed, pAsa4c and pAhD4-1 each belong to separate entry exclusion groups (EexE and EexD, respectively) distinct from the one encompassing IncC and IncA plasmids (EexC) [9]; yet, all belong to the same surface exclusion group. These observations suggest IncC, IncA and ACL plasmids form a two-tier community, exerting surface and entry exclusion against closely related peers but only surface exclusion against a broader range of distantly related conjugative elements.

This work has identified orphan surface exclusion factors borne by IMEs. IE*Ave*Tha1-like elements belong to the MGI*Vch*Hai6 family of IncC-mobilizable IMEs [4,5], whereas IE*Vch*Aut1-like elements constitute a new family of IMEs likely to be mobilizable by ACL plasmids (Fig 3C). We showed that $sfx_{Aut1}$ of IE*Vch*Aut1 is functional (Fig 4C). Since these IMEs lack a *traN* gene, we propose they engage in surface exclusion as a defensive mechanism, forcing their host to favour mating partners that could enhance both their stability and transmissibility. Hence, these IMEs could modulate the circulation of conjugative elements in bacterial populations without affecting whether surface exclusion from a potential recipient will target transfer mediated by their helper element. Such behaviour would contrast with the way SGI1 manipulates the exclusion system of its helper IncC plasmid. SGI1, which is not known to exert exclusion, evades IncC entry exclusion by substituting TraG in the mating apparatus with a distant VirB6 homolog [11]. Manipulation of exclusion by mobilizable elements is not well known. While homologous ColE1 mobilizable plasmids were reported to inhibit each other's transfer mildly, this phenotype was attributable to the accessory *mob* gene *mbeD* with no identified target in the donor [20,46,47]. It thus remains unclear whether exclusion is at play in that case. The putative IMEs we reported here may be the first instance of exclusion exerted by mobilizable elements. The close relationship between Sfx proteins encoded by IE*Ave*Tha1 and IE*Vch*Aut1 with those tested in this study ($Sfx_{94}$, $Sfx_{AhD4-1}$, and $Sfx_{Asa4c}$) suggests they belong to the same exclusion group (Figs 3 and S6). This activity has been corroborated by experimental evidence for $Sfx_{Aut1}$ (Figs 4C and S5), establishing surface exclusion as an actual barrier to the circulation of mobile genetic elements on par with other bacterial defense mechanisms instead of being limited to preventing redundant transfer and, potentially, mediating mating pair separation [13,39]. Surprisingly, none of the IMEs identified by the TraN search, including SGI1, appears in the Sfx tree, suggesting they do not encode a surface exclusion factor related to $Sfx_{94}$. Conversely, IMEs identified by the Sfx search are ostensibly absent from the TraN tree as they lack the components of the conjugative apparatus. This observation suggests different survival strategies. Whereas SGI1 maximizes its own dissemination by remodelling the mating apparatus to improve transfer and bypass entry exclusion [11], IE*Ave*Tha1 and IE*Vch*Aut1 may ensure persistence by hindering via surface exclusion the entry of destabilizing, excision-inducing ACL plasmids. Although the mobilization mechanism of IE*Vch*Aut1 remains unknown, its gene content and the presence of AcaCD binding motifs suggest mechanistic parallels with that of SGI1, albeit using totally unrelated mobilization factors. In $ACL^+$ cells, AcaCD likely promotes IE*Vch*Aut1's excision through the expression of *rdfM* [38]. A predicted homolog of the replication protein P from bacteriophage lambda suggests the excised IME is replicative [48]. IME DNA transfer seems to rely on the IME-encoded TraI, a predicted relaxase of the $MOB_F$ family [49]. While SGI1 encodes its own replication and DNA processing functions [10,50,51], IE*Vch*Aut1 additionally encodes a type IV coupling protein, presumably making it independent from the IncC-encoded TraD and DtrJ (also known as TraJ) [5,52].

IncHI1 EexB and IncF TraT are two other known surface exclusion factors. While EexB seems to have a dual role as an exclusion factor and exclusion target, TraT's mechanism remains elusive as exclusion specificity could not be attributed to direct interaction with pilin or indirect interaction with the putative adhesin TraN through OmpA [15,27,28,30]. Our

results demonstate that Sfx is only required in the recipient to enable exclusion (Fig 4A), ruling out the self-targeting of Sfx. Instead, we found that Sfx exclusion can be neutralized by substituting TraN in the donor with TraN from another surface exclusion group (Fig 4B and 4C). Distinct exclusion groups within a given family were previously reported in IncI and IncF plasmids and SXT/R391 ICEs [16,23,53–55].

Although a direct interaction between Eex and TraG has been suggested in SXT/R391 ICEs and likely extends to other VirB6-targeting entry exclusion factors [26], direct contact between Sfx and TraN is unlikely for surface exclusion. In fact, the binding of outer membrane proteins on donor and recipient cells is hard to reconcile with the idea that surface exclusion physically prevents the formation or disrupts mating aggregates [14]. Instead, surface exclusion factors have been proposed to act by concealing outer membrane receptors involved in mating aggregation [29]. According to Riede and Eschbach, TraT of F directly interacts with the major outer membrane protein OmpA as TraT inhibits the OmpA-specific phage K3 [29]. Yet, despite ample evidence that $TraN_F$ interacts with OmpA during conjugation, it does not seem to determine exclusion specificity as $TraN_F$ substitution with $TraN_{R100-1}$ does not counter exclusion by TraT of F [30–32].

We found that $sfx_{94}$ is constitutively expressed independently of AcaCD at a much higher level than the entry exclusion factor $eexC$ or the adhesin gene $traN$ when donor cells are in pure culture (Fig 2C and 2D). The high expression of $sfx$ relative to $eexC$, despite similar levels of exclusion exerted by the two genes (Figs 1D and 2D), is consistent with an interaction with a highly abundant surface protein. While few EexC proteins might suffice to target TraG in one mating pore during conjugation, an abundance of Sfx proteins is likely required to conceal most, if not all, TraN receptors on the recipient surface at any time. In pure cultures, expression of $traN$ remained low, suggesting that the level of Sfx could be sufficient to quench spurious donor-donor mating (Fig 2D). Unlike TraT of F, we have identified TraN as the specificity determinant for Sfx-mediated exclusion between ACL plasmids. Thus, we propose that $TraN_{94}$ and $TraN_{AQU1}$ recognize distinct receptors on the recipient surface and that Sfx prevents mating pair stabilization by specific binding to the receptor of its cognate TraN. Further investigation will be required to identify the receptors involved.

## Materials and methods

### Bacterial strains and media

The bacterial strains and plasmids used in this study are described in Table 2. Bacterial strains were grown at 37°C on Luria-Bertani (LB) agar or with agitation in LB broth and maintained at −75°C in LB broth containing 20% (vol/vol) glycerol. Antibiotics were used at the following concentrations: ampicillin (Ap), 100 μg/ml; kanamycin (Kn), 50 μg/ml; spectinomycin (Sp), 50 μg/ml; rifampicin (Rf), 50 μg/ml; chloramphenicol (Cm), 20 μg/ml and nalidixic acid (Nx), 40 μg/ml. Media were supplemented with 0.02% L-arabinose or 0.1 mM isopropyl β-D-1-thiogalactopyranoside (IPTG) to induce gene expression from pBAD30 and pAH56 constructs, respectively.

### Plasmid and strain construction

Plasmid DNA was prepared using the QIAprep Spin Miniprep kit (Qiagen), and genomic DNA was isolated with the QIAamp DNA mini kit (Qiagen) as recommended by the manufacturer. PCR products were purified using the PCR purification kit (Qiagen). All molecular biology manipulations were carried out by standard procedures following the Current Protocols in Molecular Biology [70]. The oligonucleotides used in this study are described in S1 Table. Deletions were constructed using the one-step chromosomal gene inactivation technique (λ

**Table 2. Strains and plasmids used in this study.**

| Strain, plasmid or element | Relevant genotype or phenotype | Reference |
|---|---|---|
| *Escherichia coli* | | |
| DH5α | F⁻φ80*lacZ*ΔM15 Δ(*lacZYA-argF*)U169 *recA1 endA1 hsdR17*(r$_K^-$, m$_K^+$) *phoA supE44* λ⁻*thi-1 gyrA96 relA1* | [56] |
| BW25113 | F–Δ(*araD-araB*)567 Δ*lacZ4787*(::*rrnB-3*) λ–*rph-1* Δ(*rhaD-rhaB*)568 *hsdR514* | [57] |
| GG56 | Nx$^r$ derivative of BW25113 (Nx) | [58] |
| KH95 | Rf$^r$ derivative of BW25113 (Rf) | [10] |
| MFD*pir* | MG1655 RP4-2-Tc::[ΔMu1::*aac(3)IV*-Δ*aphA*-Δ*nic35*-ΔMu2::*zeo*] Δ*dapA*::(*erm-pir*) Δ*recA* | [59] |
| VB111 | Nx$^r$ derivative of MG1655 (Rf) | [60] |
| VB112 | Rf$^r$ derivative of MG1655 (Rf) | [60] |
| CAG18439 | MG1655 *lacZU118 lacI42*::Tn*10* (Tc) | [61] |
| *Vibrio cholerae* | | |
| A12JL5W90 | non-O1/non-O139 isolated from a large Austrian lake, 2012 | [62] |
| Plasmids | | |
| pVCR94 | IncC conjugative plasmid, *V. cholerae* O1 El Tor (Su Tm Cm Ap Tc Sm) | [63] |
| pVCR94$^{Sp}$ | Sp$^r$ derivative of pVCR94 (pVCR94ΔX2) (Su Sp) | [11] |
| pVCR94$^{Kn}$ | Kn$^r$ derivative of pVCR94 (pVCR94ΔX3) (Su Kn) | [11] |
| pVCR94$^{Kn}$ Δ*eexC*::*cat* | *eex* deletion mutant of pVCR94$^{Kn}$ (Su Kn Cm) | This study |
| pFG036 | *ori*$_{pMB1}$, *cI857* (Ts) repressor, *tetM* (Tc) | Addgene #137996 |
| pFG051 | *ori*$_{R6K}$, Tn*5 tnp* under λP$_L$ promoter, *oriT*$_{RP4}$, Tn*5d-aadA7* (Sp) | Addgene #137997 |
| pVCR94$^{Sp}$ Δ*018–019* | *vcrx018-vcrx019* deletion mutant of pVCR94$^{Sp}$ (Su Sp) | This study |
| pVCR94$^{Sp}$ Δ*062* | *vcrx062* deletion mutant of pVCR94$^{Sp}$ (Su Sp) | This study |
| pVCR94$^{Sp}$ Δ*107–108* | *vcrx107-vcrx108* deletion mutant of pVCR94$^{Sp}$ (Su Sp) | This study |
| pVCR94$^{Sp}$ Δ*111* | *vcrx111* deletion mutant of pVCR94$^{Sp}$ (Su Sp) | This study |
| pVCR94$^{Sp}$ Δ*137* | *vcrx137* deletion mutant of pVCR94$^{Sp}$ (Su Sp) | This study |
| pVCR94$^{Sp}$ Δ*acr2* | *vcrx150* deletion mutant of pVCR94$^{Sp}$ (Su Sp) | [7] |
| pVCR94$^{Sp}$ Δ*eexC* | *vcrx145* deletion mutant of pVCR94$^{Sp}$ (Su Sp) | [9] |
| pVCR94$^{Sp}$ Δ*sfx* | *vcrx085* deletion mutant of pVCR94$^{Sp}$ (Su Sp) | This study |
| pVCR94$^{Sp}$ Δ*eexC*Δ*sfx* | *vcrx145-vcrx085* double deletion mutant of pVCR94$^{Sp}$ (Su Sp Cm) | This study |
| pVCR94$^{Kn}$ Δ*acaCD* | *acaCD* deletion mutant of pVCR94$^{Kn}$ (Su Kn) | This study |
| pClo | CloDF13::TnAΔEcoRV (pSU4628) (Ap) | [64] |
| p*acaCD* | pBAD30::*acaCD* (Ap) | [7] |
| pVCR94$^{Sp}$ Δ*traN* | *traN* deletion mutant of pVCR94$^{Sp}$ (Su Sp) | [11] |
| pVCR94$^{Kn}$ Δ(*087-traG*) | *vcrx87-traG* deletion mutant of pVCR94$^{Kn}$ (Su Kn) | This study |
| pAsa4c | Cm$^r$ conjugative plasmid from *A. salmonicida* JF2267 (Cm) | [41] |
| p*traN*$_{94}$ | pBAD30::*traN*$_{pVCR94}$ (Ap) | [11] |
| p*traN*$_{Asa4c}$ | pBAD30::*traN*$_{pAsa4c}$ (Ap) | This study |
| p*traN*$_{AhD4-1}$ | pBAD30::*traN*$_{pAhD4-1}$ (Ap) | This study |
| p*traN*$_{AQU1}$ | pBAD30::*traN*$_{pAQU1}$ (Ap) | This study |
| p*sfx*$_{94}$ | pAH56::*sfx*$_{pVCR94}$ (Kn) | This study |
| p*sfx*$_{Asa4c}$ | pAH56::*sfx*$_{pAsa4c}$ (Kn) | This study |
| p*sfx*$_{AhD4-1}$ | pAH56::*sfx*$_{pAhD4-1}$ (Kn) | This study |
| p*sfx*$_{AQU1}$ | pAH56::*sfx*$_{pAQU1}$ (Kn) | This study |
| p*sfx*$_{Aut1}$ | pAH56::*sfx*$_{IEVchAut1}$ (Kn) | This study |
| p*eexC* | pAH56::*eex*$_{pVCR94}$ (Kn) | This study |
| p*sfx*$^{3×Flag}$ | pAH56::*sfx*$_{pVCR94}$$^{3×Flag}$ (Kn) | This study |
| p*acaDC*$^{3×Flag}$ | pAH56::*acaDC*$^{3×Flag}$ (Kn) | [7] |
| pVCR94$^{Kn}$ *eexC*'-'*lacZ* | Translational fusion of the 8$^{th}$ codon of *lacZ* at the 3$^{rd}$ codon of *vcrx145* (Su Kn) | This study |
| pVCR94$^{Kn}$ *sfx*'-'*lacZ* | Translational fusion of the 8$^{th}$ codon of *lacZ* at the 3$^{rd}$ codon of *vcrx085* (Su Kn) | This study |

*(Continued)*

**Table 2.** (Continued)

| Strain, plasmid or element | Relevant genotype or phenotype | Reference |
|---|---|---|
| pVCR94$^{Kn}$ traN'-'lacZ | Translational fusion of the 8$^{th}$ codon of lacZ at the 3$^{rd}$ codon traN (Su Kn) | This study |
| pVCR94$^{KnΔCD}$ eexC'-'lacZ | acaCD deletion mutant of pVCR94$^{Kn}$ eexC'-'lacZ (Su Kn Cm) | This study |
| pVCR94$^{KnΔCD}$ sfx'-'lacZ | acaCD deletion mutant of pVCR94$^{Kn}$ sfx'-'lacZ (Su Kn Cm) | This study |
| pVCR94$^{KnΔCD}$ traN'-'lacZ | acaCD deletion mutant of pVCR94$^{Kn}$ traN'-'lacZ (Su Kn Cm) | This study |
| peexC-sfx | pBeloBAC11::(sfx-eexC) from pVCR94$^{Kn}$ Δ(087-traG) (Cm) | This study |
| pBeloBAC11 | Single-copy vector derived from the F plasmid (Cm) | New England Biolabs |
| pAH56 | oriV$_{R6Kγ}$ attP$_λ$ lacI P$_{tac}$ (Kn) | [65] |
| pBAD30 | ori$_{p15A}$ bla araC P$_{BAD}$ (Ap) | [66] |
| pINT-ts | ori$_{R101}$ cI857 λp$_R$-int$_λ$ (ts, Ap) | [65] |
| pVI42b | pVI36 BamHI::P$_{lac}$-lacZ (Sp) | [67] |
| pKD3 | cat template for one-step chromosomal gene inactivation (Cm) | [57] |
| pKD4 | aph template for one-step chromosomal gene inactivation (Kn) | [57] |
| pCP20 | Flp recombinase thermo-inducible encoding plasmid (ts, Ap Cm) | [68] |
| pSIM6 | λ Red recombination thermo-inducible encoding plasmid (ts, Ap) | [69] |

Ap, ampicillin; Cm, chloramphenicol; Kn, kanamycin; Nx, nalidixic acid; Rf, rifampicin; Sm, streptomycin; Sp, Spectinomycin; Su, sulfamethoxazole; Tc, tetracycline; Tm, trimethoprim; ts, thermosensitive

Red-mediated mutagenesis) with pKD3 and pKD4 as templates for antibiotic resistance cassettes [57]. The deletions of vcrx018-vcrx019, vcrx062, vcrx085, vcrx107-vcrx108, vcrx111, vcrx137 and eexC in pVCR94$^{Sp}$ or pVCR94$^{Kn}$ were done using primer pairs 94delvcrx018.for/94delvcrx019.rev, 94delvcrx062.for/94delvcrx062.rev, 94delvcrx085.for/94delvcrx085.rev, 94delvcrx107.for/94delvcrx108.rev, 94delvcrx111.for/94delvcrx111.rev, 94delvcrx137.for/94delvcrx137.rev and 94deI145.for/94deI145.rev, respectively, and pKD3 or pKD4 as the template. Deletion of vcrx085 in pVCR94$^{Sp}$ ΔeexC was done using primer pair 94delsfx.for/94delsfx.rev and pKD3 as the template. The λRed recombination system was expressed using pSIM6. When appropriate, resistance cassettes were excised from the resulting constructions using the Flp-encoding plasmids pCP20. All deletions were validated by antibiotic profiling and PCR.

The eexC'-'lacZ, sfx'-'lacZ, and traN'-'lacZ translational fusions were introduced into pVCR94$^{Kn}$ using primer pairs 94eex-lacZ.f/94eex-lacZ.r, 94sfx-lacZ.f/94sfx-lacZ.r, and 94traN-lacZ.f/94traN-lacZ.r, and pVI42b as the template. In all cases, the third codon of the gene of interest was fused to the eighth codon of lacZ. The deletion of acaCD in pVCR94$^{Kn}$, pVCR94$^{Kn}$ eexC'-'lacZ, pVCR94$^{Kn}$ sfx'-'lacZ, and pVCR94$^{Kn}$ traN'-'lacZ was done using primer pair 94DelacaD.for/94DelacaC.rev and pKD3 as the template.

pVCR94 and pAsa4c served as templates to amplify their respective genes of interest. sfx$_{AhD4-1}$, sfx$_{AQU1}$, traN$_{AhD4-1}$ and traN$_{AQU1}$ were chemically synthesized (Bio Basic, Markham). sfx$_{94}$, sfx$_{Asa4c}$, sfx$_{AhD4-1}$, sfx$_{AQU1}$ and eexC were amplified using primer pairs 94sfxNdeI.f/94sfxSalI.r, Asa4sfxNdeI.f/Asa4sfxSalI.r, AhD4sfxNdeI.f/AhD4sfxSalI.r, AQU1sfxNdeI.f/AQU1sfxSalI.r, and eexCNdeI.f/eexCSalI.r, respectively. sfx$_{Aut1}$ was amplified using primer pair Aut1sfxNdeI.f/Aut1sfxSalI.r and the genomic DNA of V. cholerae A12JL5W90 as the template. Amplicons were digested with NdeI and SalI and cloned into NdeI/SalI-digested pAH56. psfx$^{3xFlag}$ was constructed from psfx$_{94}$ using primer pair pAH56_3xFlag.F/94sfx_3xFlag.R and the Q5 Site-Directed Mutagenesis Kit (New England Biolabs) according to the manufacturer's instructions. The resulting constructs were single-copy integrated into the attB$_λ$ chromosomal site of CAG18439 using pINT-ts.

To assemble a DNA fragment encompassing *sfx*$_{94}$ and *eexC* alongside their native promoters, the region spanning from *acaB* to *traG* in pVCR94$^{Kn}$ was deleted using the one-step chromosomal gene inactivation technique with primer pair 94del86acaB.for/94del144traG.rev and pKD3 as the template. After the excision of the resistance cassette using pCP20, the region of interest was amplified using primer pair 94eexBamHI.for/94sfxHindIII.rev. The amplicon was digested with BamHI and HindIII and cloned into a BamHI/HindIII-digested pBeloBAC11.

*traN*$_{Asa4c}$, *traN*$_{AhD4-1}$, *traN*$_{AQU1}$ were amplified using primer pairs Asa4traNEcoRI.f/Asa4-traNSalI.r, AhD4traNEcoRI.for/AhD4traNEcoRI.rev and AQU1traNEcoRI.for/AQU1traNEcoRI.rev, respectively. Amplicons were digested with EcoRI or EcoRI/SalI and cloned into EcoRI- or EcoRI/SalI-digested pBAD30.

All constructs were verified by PCR and DNA sequencing at the Plateforme de Séquençage et de Génotypage du Centre de Recherche du CHUL (Québec, QC, Canada).

## Bacterial conjugation and exclusion assays

Bacteria were grown for 16 h in LB broth with the appropriate antibiotics. Mating assays were carried out by mixing 100 μl of donor and recipient cells. Cells were pelleted by centrifugation, then washed once in 1 volume of LB broth and resuspended in 1/20 volume of LB broth. Bacterial mixtures were incubated for 6 h on LB agar plates at 37˚C to allow conjugation. Serial dilutions were then plated on selective LB-agar plates with appropriate antibiotics to discriminate between donor, recipient and transconjugant CFUs. Transfer frequencies were calculated by dividing the number of transconjugant CFUs by the number of donor CFUs. Exclusion indices were calculated as the ratio of an element's transfer frequency toward an empty recipient to its transfer frequency toward the tested recipient.

## Transposon-directed insertion sequencing (TraDIS)

A conjugation-assisted random transposon mutagenesis experiment was performed on *E. coli* GG56 bearing pVCR94$^{Kn}$ Δ*eexC*::*cat*. The transposition system was composed of *E. coli* MFD*pir* carrying pFG036 (a plasmid coding for the thermosensitive cI857 transcriptional repressor) and pFG051 (a Pi-dependent RP4-mobilizable plasmid coding for the Tn*5* transposition machinery repressed by cI857 and carrying a mini-Tn*5* (Sp) transposon). This diaminopimelate (DAP)-auxotrophic strain contains a chromosomal RP4 conjugative machinery and expresses the Pi protein required for pFG051 replication. The TraDIS experiment was performed in several successive steps. First, pFG051 was transferred by conjugation from MFD*pir* to GG56 bearing pVCR94$^{Kn}$ Δ*eexC*::*cat* in a 2-h mating experiment at 30˚C on LB agar plates supplemented with DAP in duplicates. Once in the recipient strain that lacks cI, the constitutively expressed Tn*5* machinery of pFG051 mediates random mini-Tn*5* (Sp) insertions in the genome. The mating mixture was then entirely spread onto 40 large LB agar plates (150 mm) supplemented with Cm, Kn, Nx, and Sp. Plates were incubated until near confluence (40k to 60k CFUs) to select clones carrying mini-Tn*5* (Sp) insertions. After overnight incubation at 37˚C, Cm Nx Kn Sp-resistant colonies were collected using a cell scraper and resuspended in LB broth. The collected sample, designated as the 'input library' was washed, then resuspended in 4.5 ml of LB broth and cryopreserved. The total DNA of a 1.5 ml aliquot of the input library was extracted for sequencing. Another 1 ml aliquot of input library was used to inoculate 50 ml of LB broth supplemented with Cm, Kn, Nx and Sp, which was then grown overnight at 37˚C. The resulting culture was used as the recipient in a mating assay, mixed in equal volumes with *E. coli* KH95 carrying pVCR94$^{Kn}$ and pClo used as the donor. After 2-h incubation at 37˚C, mating mixtures were spread onto 20 large LB agar plates (150 mm) supplemented with Nx, Kn, Sp, Cm, and Ap to select for mini-Tn*5* (Sp) insertions that allowed the entry and replication of pClo into the

recipients. After overnight incubation at 37˚C, Nx, Kn, Sp, Cm, and Ap-resistant colonies were collected and subsequently resuspended in LB broth, washed and resuspended in 4.5 ml of LB broth and cryopreserved. These samples were designated as 'output libraries'. The total DNA of a 1.5 ml aliquot of the output libraries was extracted and used for sequencing.

### Preparation of TraDIS libraries and Illumina sequencing

For each library, a 1.5 ml frozen stock aliquot was thawed on ice for 15 min and used to prepare sequencing libraries as described previously [71]. Mutant libraries were then pooled and sequenced by Illumina using the NextSeq 500/550 High Output Kit v2 at the RNomics platform of the Laboratoire de Génomique Fonctionnelle de l'Université de Sherbrooke (https://rnomics.med.usherbrooke.ca) (Sherbrooke, QC, Canada). The transposon data analysis was carried out as described previously [71].

### Fractionation of cellular proteins

Cell fractionation was carried out using a protocol derived from Sandrini *et al.* [72]. Briefly, *E. coli* DH5α *λpir* bearing p*acaDC*$^{3×Flag}$ or p*sfx*$^{3xFlag}$ was grown overnight with or without 0.1 mM IPTG. The cells were pelleted at 3,900 *g* for 20 min at 4˚C. Cell pellets were washed twice with 10 mM Tris buffer (pH 7.5), resuspended in 10 ml of the same buffer, and frozen for 2 h at -80˚C. Samples were lysed by ultrasonication (Qsonica Q125) in an ice bath using five cycles of 30 s each at 70% amplitude, followed by 30 s of cooling. The lysates were treated with DNAse I (1 mg/ml). Cell debris were removed by centrifugation (3,900 *g* for 30 min at 4˚C). The proteins in the supernatant were separated by ultracentrifugation (185,000 *g* for 30 min at 4˚C). The supernatant (cytoplasmic proteins) was stored at -20˚C. The pellet (total membrane proteins) was washed three times with 10 mM Tris buffer (pH 7.5), resuspended in 10 mM Tris buffer with 2% (v/v) Triton X-100, and incubated for 30 min at room temperature. The mixture was centrifuged at 185,000 *g* for 30 min at 4˚C. The supernatant (inner membrane proteins) was concentrated using Pierce Protein Concentrators PES 3K, following the manufacturer's instructions (Thermo Scientific, cat.# 88515) and stored at -20˚C. The pellet (outer membrane proteins) was washed three times with 10 mM Tris buffer (pH 7.5) and stored at -20˚C.

### SDS-page and immunoblot analysis

Bradford quantification was used to determine the quantity of fractionated proteins and normalize protein samples. Each normalized fraction was resuspended in NuPAGE LDS Sample Buffer (Invitrogen) supplemented with NuPAGE Sample Reducing agent (Invitrogen), incubated at 70˚C for 10 min, and separated by 12% SDS-PAGE (120 mV, 2 h). Proteins were detected by silver staining (Pierce Silver Stain Kit) following the manufacturer's instructions (Thermo Scientific, cat.# 24612) or transferred onto a polyvinylidene fluoride (PVDF) membrane. Western blotting was performed according to standard procedures. The membranes were probed using 1:300 rabbit OMPC polyclonal primary antibody (ThermoFisher, cat # BS-20213R) or 1:3000 rabbit anti-FLAG primary antibody (Sigma-Aldrich, cat. # F7425), then with 1:3000 HRP goat anti-rabbit IgG HRP-conjugated secondary antibody (Invitrogen, cat. # G-21234). Detection was carried out with the Clarity western ECL substrate (BioRad, cat.# 170–5061) using the ChemiDoc MP imaging system (BioRad).

### Molecular biology

Plasmid DNA was extracted using the EZ-10 Spin Column Plasmid DNA Minipreps Kit (Bio Basic) following the manufacturer's instructions. Enzymes used in this study were purchased

from New England Biolabs. PCR assays were performed with primers listed in S1 Table. PCR conditions were as follows: (i) 3 min at 94°C; (ii) 30 cycles of 30 s at 94°C, 30 s at the appropriate annealing temperature, and 1 minute/kb at 68°C; and (iii) 5 min at 68°C. When required, the resulting products were purified using the EZ-10 Spin Column PCR Products Purification Kit (Bio Basic) following the manufacturer's instructions. *E. coli* strains were transformed by electroporation as described previously [73] in a Bio-Rad Gene Pulser Xcell device set at 25 μF, 200 V and 1.8 kV using 1-mm gap electroporation cuvettes.

## RNA extraction and cDNA synthesis

RNA extractions were performed as follows. *E. coli* GG56 containing pVCR94$^{Kn}$ with or without p*acaCD* was grown at 37°C for 16 h in LB broth containing the appropriate antibiotics. The cultures were diluted 1:200 in fresh medium containing the appropriate antibiotics and grown to an $OD_{600}$ of 0.2 before being diluted 1:10 again in fresh medium containing the appropriate antibiotics and supplemented with 0.02% arabinose when needed. After a 2-h incubation period, 4 ml of the cultures were used for total RNA extraction using the Direct-zol RNA extraction kit (Zymo Research) and TRI Reagent (Sigma-Aldrich) according to the manufacturer's instructions. Once purified, the RNA samples were treated using 2 units of DNase I (New England Biolabs) according to the manufacturer's instructions to eliminate residual gDNA. cDNA was synthesized from 0.5 ng of RNA and 1 pmol of gene-specific primer 85RT (Integrated DNA Technologies) using the reverse transcriptase SuperScript IV (Invitrogen), according to the manufacturer's instructions. Control reactions without reverse transcriptase treatment ('RNA') were performed for each sample. PCR reactions aiming at amplifying $sfx_{94}$ and *acaD* were carried out using 50 pg of RNA or the corresponding amount of cDNA, or 0.5 ng of gDNA as the template, and primer pairs sfxF/sfxR and 086F2/086R2, respectively.

## β-galactosidase assays

Quantitative β-galactosidase assays were performed with 2-nitrophenyl-β-D-galactopyranoside (ONPG) according to a protocol adapted from Miller [74]. After overnight incubation at 37°C with appropriate antibiotics, cultures were diluted 1:100 in 50ml LB broth supplemented with 50 μg/ml ampicillin and grown until an $OD_{600}$ of 0.2 was reached. Three series of 1/10 dilutions were then prepared in total volumes of 5 ml LB broth supplemented with 50 μg/ml ampicillin with or without 0.2% arabinose and incubated for 2 hours at 37°C.

## Statistical analyses

Numerical data presented in graphs are available in S1 File. All analyses were performed using R Statistical Software (v4.2.1) and RStudio (v2022.12.0.252) [75]. We assumed a normal distribution of the frequencies of transconjugant formation and tested the homogeneity of variance of the $log_{10}$-transformed frequencies using Bartlett's tests. Dunnett and Tukey-Kramer tests and compact letter display were performed via the multcomp R package (v1.4–20) [76]. Details of statistical tests are provided in S2 File. Graphics were rendered via the ggplot2 R package (v3.3.6) [77]. All figures were prepared using Inkscape 0.92 (https://inkscape.org/).

## Phylogenetic analyses

The primary sequence of Sfx$_{94}$ and TraN$_{94}$ homologs were obtained using the NCBI blastp algorithm [78] against the nr/nt database restricted to Gammaproteobacteria (taxid: 1236). Primary sequences sharing less than 45% identity and under 85% minimum coverage were filtered out of subsequent analyses. The distantly related Sfx$_{AQU1}$ from pAQU1 was added

manually to the Sfx dataset as an outgroup. The datasets were clustered with CD-HIT [79] to the best cluster that met the 0.90 identity cut-off before alignment. Where applicable, representative sequences ("seeds") were manually substituted with sequences of interest within the same clusters for downstream analyses. The resulting datasets were then aligned with MUS-CLE [80], and poorly aligned regions were discarded from the resulting amino acid alignments using TrimAl v1.3 software with the automated heuristic approach [81]. Evolutionary analyses were performed within MEGA11 (v 11.0.13) [82] using the Maximum Likelihood method (PhyML) [83] and the Le and Gascuel matrix-based model [84] with gamma distribution (LG + G). Initial tree(s) for the heuristic search were obtained automatically by applying Neighbor-Join and BioNJ algorithms to a matrix of pairwise distances estimated using a JTT model and then selecting the topology with superior log likelihood value. Percent identity values were retrieved from the MUSCLE output [80].

### Bioinformatic predictions

The prediction of signal peptides and cleavage sites in $Sfx_{94}$ and $TraN_{94}$ was performed using SignalP 6.0 with the slow model mode against other organisms [85]. Terminators were predicted on both strands using ARNold [86,87]. *oriT* loci and mobilization proteins in putative IMEs were predicted using oriTfinder [88].

### Supporting information

**S1 Fig. Identification of an additional IncC exclusion factor by TraDIS.** The tracks plot the number of reads from two independent replicates as a function of position in pVCR94$^{Kn}$ $\Delta eexC$::*cat* for both the input and output libraries. ORFs with similar functions are colour-coded as indicated in the figure. This figure was created using the UCSC Genome Browser (http://genome.ucsc.edu) [89].
(PDF)

**S2 Fig. Impact of exclusion on the fitness of IncC$^+$ cells.** Colony forming units (CFUs) were obtained from mating assays conducted in Fig 1D. Crossbars show the mean and standard error of the mean of three independent experiments. One-way ANOVA (p = 3.9e-6) with a Tukey-Kramer post-test was used on the $\log_{10}$-transformed values to compare the means. Statistical significance (S2 File) is shown as a compact letter display for pairwise comparisons where means grouped under identical letters are not statistically different.
(PDF)

**S3 Fig. Sfx is a predicted lipoprotein.** (A) Schematic representation of the predicted translation products of $sfx_{94}$ and $traN_{94}$ and their predicted signal peptide sequences. (B) Predicted Rho-independent terminators in the proximity of *sfx* in the IncC plasmid pVCR94. Terminator sequences are provided with the stems and loops in blue and red, respectively. Their position is represented by hairpins on the schematic representation of the region. Predicted ΔG values are indicated in kcal/mol.
(PDF)

**S4 Fig. Functionality of 3xFLAG-labelled Sfx$_{94}$.** pVCR94$^{Sp}$ was transferred from *E. coli* VB111 (Nx$^r$) into *E. coli* DH5α λ*pir* bearing the indicated pAH56 derivatives (Kn$^r$) expressing $sfx_{94}$ or $sfx_{94}$-3xFlag in the presence or absence of IPTG. Transconjugants containing pVCR94$^{Sp}$ were selected as the Kn$^r$ Sp$^r$ colonies. Crossbars show the mean and standard error of the mean of three independent experiments. One-way ANOVA (p = 7.3e-8) with a Tukey-Kramer post-test was used on the $\log_{10}$-transformed values to compare the means for the

IPTG-induced condition.
(PDF)

**S5 Fig. TraN directs surface exclusion specificity.** *E. coli* GG56 (Nx$^r$) containing pVCR94$^{Sp}$ Δ*traN* and expressing the indicated *traN* variants from pBAD30 (Ap$^r$) served as the donor, and *E. coli* CAG18439 (Tc$^r$) expressing the indicated *sfx* variants from a single-copy, chromosomally integrated pAH56 (Kn$^r$) was used as the recipient. Transconjugants containing pVCR94$^{Sp}$ Δ*traN* were selected as the Tc$^r$ Sp$^r$ colonies. Crossbars show the mean and standard error of the mean of three independent experiments. One-way ANOVA (*traN* pVCR94, p = 2.21e-8; *traN* pAsa4c, p = 4.84e-10; *traN* pAhD4-1, p = 2.20e-8; *traN* pAQU1, p = 1.81e-9) with a Tukey-Kramer post-test was used on the log$_{10}$-transformed values to compare the means. Exclusion indices, shown at the bottom of each crossbar, are calculated by dividing the frequency of transfer of pVCR94$^{Sp}$ Δ*traN* to the empty recipient divided by the transfer frequency to the recipient expressing the indicated *sfx* variant. Statistical significance (S2 File) is shown as a compact letter display for pairwise comparisons where means grouped under identical letters are not statistically different.
(PDF)

**S6 Fig. Sequence identity heatmap showing pairwise percentage identity between 18 selected Sfx homologs.** Panels are shaded as indicated in the figure according to the identity calculated for each pair.
(PDF)

**S1 Table. Primers used in this study.**
(DOCX)

**S2 Table. Predicted AcaCD binding sites.**
(DOCX)

**S1 File. Complete data sets.**
(XLSX)

**S2 File. Statistical analyses.**
(XLSX)

**S3 File. Sfx and TraN homologs.**
(XLSX)

## Acknowledgments

We are grateful to Emilie Boulerne, Colin Poirier, Simon Jeanneau, and Frédéric Grenier for technical assistance. We thank Dr. Alexander Kirschner, Medical University of Vienna, for the kind gift of *V. cholerae* A12JL5W90.

## Author Contributions

**Conceptualization:** Nicolas Rivard, Malika Humbert, Kévin T. Huguet, Vincent Burrus.

**Data curation:** Nicolas Rivard, César Pérez Bucio, Vincent Burrus.

**Formal analysis:** Nicolas Rivard, Vincent Burrus.

**Funding acquisition:** Nicolas Rivard, César Pérez Bucio, Vincent Burrus.

**Investigation:** Nicolas Rivard, Malika Humbert, Kévin T. Huguet, Aurélien Fauconnier, César Pérez Bucio, Eve Quirion.

**Methodology:** Nicolas Rivard, Malika Humbert, Kévin T. Huguet, Aurélien Fauconnier, Vincent Burrus.

**Project administration:** Vincent Burrus.

**Supervision:** Vincent Burrus.

**Validation:** Vincent Burrus.

**Visualization:** Nicolas Rivard, Aurélien Fauconnier.

**Writing – original draft:** Nicolas Rivard, Aurélien Fauconnier, Vincent Burrus.

**Writing – review & editing:** Nicolas Rivard, Malika Humbert, Kévin T. Huguet, Aurélien Fauconnier, César Pérez Bucio, Eve Quirion, Vincent Burrus.

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
