## [Decision Letter · Decision Letter 0]

2 Sep 2024

Dear Dr Burrus,

Thank you very much for submitting your Research Article entitled 'Surface exclusion of IncC conjugative plasmids and their relatives' to PLOS Genetics.

The manuscript was fully evaluated at the editorial level and by independent peer reviewers. The reviewers appreciated the attention to an important topic but identified some concerns that we ask you address in a revised manuscript.

We therefore ask you to modify the manuscript according to the review recommendations. Your revisions should address the specific points made by each reviewer.

To resubmit, log into your Editorial Manager account and select the option 'Revise Submission' in the 'Submissions Needing Revision' folder.

Yours sincerely,

Ankur B. Dalia

Academic Editor

PLOS Genetics

Sean Crosson

Section Editor

PLOS Genetics

As you will see below, all three authors are very enthusiastic about the revised manuscript. There is only one minor point that needs to be addressed. This is related to the subcellular localization experiments, which could be addressed by tempering / qualifying conclusions of the experiments already performed. Or this point can be addressed by performing additional experiments.

Reviewer's Responses to Questions

**Comments to the Authors:**

Reviewer #1: In the revised version of the manuscript titled "Surface exclusion of IncC conjugative plasmids and their relatives.", the authors have provided a detailed and thoughtful response to the concerns initially raised by this reviewer, significantly improving the manuscript's quality and clarity.

As suggested, the authors have integrated exclusion indices when required and justified when they did not. The explanation regarding the use of a log10 transformation in Figure 4C, followed by a revision to a linear scale, further clarifies the data presentation and resolves the reviewer's concerns about the low exclusion index values. The distinction between the expression of traN and sfx from their native promoters versus inducible promoters is now clearly articulated, providing a robust explanation for the observed differences in exclusion index values across the figures.

The reviewer also expressed concerns about the potential impact of exclusion system inactivation on the fitness and viability of the strains used. The authors have satisfactorily addressed this issue by presenting additional data (now included as S2 Fig and S1 File). The additional discussion of the effects of overexpressing sfx or eexC, which did show a negative impact on colony-forming units (CFU), further enhances the manuscript by providing a comprehensive view of the genetic manipulations' effects on bacterial fitness.

Minor comments raised by the reviewer have also been addressed effectively. The authors have included results for the double complementation of eex and sfx, which restores full exclusion, thereby providing a more complete picture of the exclusion mechanisms at play.

In conclusion, the authors have successfully addressed all the reviewer's concerns, making substantial improvements to the manuscript. The revisions have resulted in a more robust, clear, and comprehensive study that makes a significant contribution to the field of conjugation biology. Given these thorough and satisfactory revisions, I recommend the acceptance of this manuscript for publication.

Reviewer #2: In this revised manuscript, the authors are reporting the identification and characterization of a new exclusion factor that functions to prevent redundant conjugative transfer of plasmids of the IncA/C incompatibility groups. The authors have extensively revised their manuscript in response to the prior reviews, which in my opinion considerably strengthened and clarified many sections. The overall findings, however, are unchanged and significant. The authors identified a new exclusion factor, termed sfx, supplied some preliminary characterization of its production levels and subcellular localization, identified homologs widely among IncA/C plasmids through bioinformatics analyses, and perhaps most excitingly determined a functional relationship between Sfx and the surface adhesin TraN. I believe the authors have satisfactorily addressed the prior concerns, and I have only a few further points for them to address. I don’t think further experiments are required, with the possible exception of the fractionation experiments. Overall, the findings add significantly to our understanding of the mechanisms evolved by plasmids to suppress redundant transfer and favor dissemination to new recipient populations.

Comments:

1. L. 151. I don’t see why monitoring pClo mobilization avoids monitoring effects of chromosomal genes in the recipient. Please provide further explanation.

2. L. 192. Granted that defining the subcellular localization of a lipoprotein is challenging, I was a bit disappointed by the (lack of) depth of the analyses probing Sfx subcellular localization, and by the interpretation of the data. A strain in which Sfx was inducibly produced from a Plac promoter on a plasmid for a prolonged overnight growth period served as the starting material for the fractionation. This easily could result in artefactual overproduction and resulting membrane mislocalization or formation of inclusion bodies. Production of the tagged protein from its native promoter would avoid these potential artefacts. Additionally, other biochemical approaches such as protease shaving or immunofluorescence microscopy with antibodies against the FLAG tag might reveal an extracellular disposition. I don’t think these experiments are necessary as long as the conclusions from the present experiments are tempered – the blots suggest that Sfx is localized to both membranes with a dominant portion of the protein in the IM. Obviously, definition of Sfx localization is critical to understand its mechanism of action and nature of its functional interaction with TraN, but this easily could be the subject of another exciting manuscript.

3. L. 189. Fig. S2. The sfx and eex deletions elevate CFU counts by approx. ½ log, raising counts to higher levels than observed for the plasmid-free strain. By contrast, overexpression reduces CFU counts by 0.1 – 0.2 logs relative to the strain carrying WT plasmid. Together, these data suggest (to me) that eex or sfx expression account for the totality of the plasmid-imposed growth suppression evidenced by comparison of the plasmid-free vs plasmid-carrying strains.

4. L. 289-294. I don’t understand this – the first sentence states that the eexC and sfx deletion mutant donors transferred pClo at elevated frequencies, whereas the second sentence states that on the contrary the sfx and eex/sfx deletion mutant donors showed a modest but significant decrease in pClo transfer. The sfx deletion mutant can’t both elevate and decrease transfer (the figure shows a slight reduction).

5. Also, I’m not sure why these data ‘suggest that augmented redundant transfer between donors was detrimental to transfer toward recipients’. This experiment does not test for redundant transfer, also, what is being redundantly transferred – the pVCR94 mutant plasmids or pClo?

6. For Fig. 4A, no p values are presented, so ‘statistical significance’ is not established.

7. I did some Alphafold comparisons of TraN’s encoded by pVCR94, pAqu1 and others encoded by the IncA/C plasmids. With this limited analysis, the putative extracellular sensor domains appear to be highly structurally conserved despite the observed differences in the functionalities when Sfx subunits were paired with cognate vs noncognate TraN’s. It would be a simple extension of the manuscript to compare the Alphafold models of TraN’s from plasmids encoding similar vs distantly related Sfx’s to probe whether the extracellular sensors exhibit structural variations (which would be expected if there were to be a direct TraN – Sfx interaction, or even distinct interactions between these TraN’s and different Omps as has been reported for the F-like TraN’s). Also, since TraN’s encoded by F-like plasmids have received a lot of attention recently, it would be informative for the authors to compare the TraN’s from IncA/C with those encoded by F-like plasmids. For example, they share very little sequence relatedness and the former are much larger than the latter. Yet the Alphafold models predict similar b-sheet extensions presumably corresponding to the extracellular sensor domains.

Reviewer #3: The revised Rivard, Humbert et al manuscript is a significant piece of work describing the Sfx family of conjugative exclusion proteins found in IncA/C plasmid and other related mobile elements. The evidence showing the importance of the interaction between Sfx and a cognate TraN protein is good. This evidence includes the surprising observation that the distant TraN-AQU1 protein can complement the conjugation defect of a pVCR94 traN deletion mutant, but does not participate in the surface exclusion process with Sfx94. The differential use of surface exclusion vs entry exclusion among these mobile elements is interesting and novel.

The presentation of the data is much improved in this version of the manuscript.

I would have liked to see a complementation assay for the functionality of the tagged version of Sfx94, which was used for the localization assay.

**Have all data underlying the figures and results presented in the manuscript been provided?**

Reviewer #1: Yes

Reviewer #2: **No: **Figure 4 - I believe PLOS Genetics requires that the raw data for the conjugation experiments, e.g., numbers of donors, recipients and transconjugants, need to be presented in an excel spreadsheet in a supplemental file.

Reviewer #3: Yes

PLOS authors have the option to publish the peer review history of their article (what does this mean?). If published, this will include your full peer review and any attached files.

Reviewer #1: **Yes: **Christian Lesterlin

Reviewer #2: No

Reviewer #3: No

---

## [Editor Report · Decision Letter 1]

27 Sep 2024

Dear Dr Burrus,

We are pleased to inform you that your manuscript entitled "Surface exclusion of IncC conjugative plasmids and their relatives" has been editorially accepted for publication in PLOS Genetics. Congratulations!

Yours sincerely,

Ankur B. Dalia

Academic Editor

PLOS Genetics

Sean Crosson

Section Editor

PLOS Genetics

Comments from the reviewers (if applicable):

**Data Deposition**

http://datadryad.org/submit?journalID=pgenetics&manu=PGENETICS-D-24-00861R1

**Press Queries**

---

## [Editor Report · Acceptance letter]

3 Oct 2024

PGENETICS-D-24-00861R1 

Surface exclusion of IncC conjugative plasmids and their relatives 

Dear Dr Burrus, 

We are pleased to inform you that your manuscript entitled "Surface exclusion of IncC conjugative plasmids and their relatives" has been formally accepted for publication in PLOS Genetics! Your manuscript is now with our production department and you will be notified of the publication date in due course.

With kind regards,

Anita Estes

PLOS Genetics

On behalf of:
